# A Novel Strain of Probiotic *Leuconostoc citreum* Inhibits Infection-Causing Bacterial Pathogens

**DOI:** 10.3390/microorganisms11020469

**Published:** 2023-02-13

**Authors:** Karnan Muthusamy, Hyo-Shim Han, Ilavenil Soundharrajan, Jeong-Sung Jung, Mariadhas Valan Arasu, Ki-Choon Choi

**Affiliations:** 1Grassland and Forages Division, National Institute of Animal Science, Rural Development Administration, Cheonan 31000, Republic of Korea; 2Department of Biotechnology, Sunchon University, Suncheon 540742, Republic of Korea; 3Department of Botany and Microbiology, College of Science, King Saud University, P.O. Box 2455, Riyadh 11451, Saudi Arabia

**Keywords:** *Leuconostoc citreum*, probiotics, antimicrobial resistance, antibacterial activity

## Abstract

Infectious diseases caused by bacteria are at risk of spreading and prolonging due to antimicrobial resistance. It is, therefore, urgently necessary to develop a more effective antibiotic alternative strategy to control pathogen spread. In general, probiotics have been recommended as a substitute for antibiotics that inhibit pathogens. This study was isolated and probiotic characteristics and antibacterial bacterial efficiency against various infection-causing pathogens were determined by different in vitro methods. A 16S rRNA sequence confirmed that the isolated strains belonged to a species of *Leuconostoc citreum*. *L. citreum* KCC-57 and KCC-58 produced various extracellular enzymes and fermented different carbohydrates. There was significant tolerance for both strains under the harsh conditions of the gastrointestinal tract (GIT). In addition, *L. citreum* KCC-57 and *L. citreum* KCC-58 showed significant auto-aggregations and hydrophobicity properties that varied with incubation time. Moreover, the cell-free secondary supernatant (CFS) of *L. citreum* KCC-57 and *L. citreum* KCC-58 inhibited growth of *Enterococcus faecalis*, *Escherichia coli*, *Pseudomonas aeruginosa* and *Staphylococcus aureus.* According to a co-culture study, *L. citreum* KCC-57 and *L. citreum* KCC-58 were highly competitive for pathogen growth. *L. citreum* KCC-57 and *L. citreum* KCC-58 showed significant probiotic potential and strong antibacterial activities against different pathogens, suggesting that these strains could be used instead of antibiotics to control infectious pathogens.

## 1. Introduction 

The global population rate and microbial pathogen-associated infections have been increasing continuously for decades. Millions of lives are lost every year due to microbial infections. Antimicrobial resistance (AMR) is the primary cause of microbial infections. It is one of the biggest threats to human health in the 21st century because pathogens are becoming less susceptible to drug treatments. AMR is predicted to kill ten million people by 2050 [1,2]. According to the World Health Organization (WHO) and other researchers, AMR is an emergency issue that requires key strategies to control it globally [3,4,5]. There were 4.9 million deaths in 2019 from bacterial infections, of which 1.27 million were caused by antimicrobial resistance. At all ages, Western sub-Saharan Africa had the highest death rates due to AMR. Among the pathogens, *E. coli*, *A. baumannii, K. pneumoniae*, *P. aeruginosa* and *S. aureus* are associated with antimicrobial resistance. In total, 929,000 deaths were caused by these pathogens in 2019 and 3.57 million in 2020. Over 100,000 deaths have been associated with methicillin-resistant *S. aureus* in 2019 [6]. 

In order to control AMR pathogens, effective antibiotics must be invented. Recently, antibiotics have become less effective due to microbial resistance. Therefore, treatment has become more challenging. There is a need for emerging and alternative strategies to protect valuable life from AMR pathogens. According to recent research, probiotics are the most effective tool for controlling AMR pathogens. Probiotic bacteria such as *Lactobacillus* and *Bifidobacterium* have a variety of biological applications [7]. Lactic acid bacteria (LAB) produce a variety of secondary metabolites that are potential antimicrobials, antioxidants and other biological agents [8,9,10]. The secondary metabolites of LAB inhibit pathogen growth via multiple molecular mechanisms [11,12,13,14,15]. LAB can also prevent the attachment of pathogenic microbes to the epithelial cells by competing with pathogens and reducing colonization of pathogens, thereby reducing infection [16,17,18]. Therefore, we intended to use *Leuconostoc citreum* from rice plants to control various pathogens in this regard. *L. citreum* is a dextran-producing species and is believed to have significant probiotic potential due to its putative plasmid-encoded cell wall anchored protein, which contains five mucus-binding domains essential for colonizing the digestive system. As well as producing various antimicrobial peptides [19,20,21], it is also capable of killing cancer cells [22]. Hence, in the present study, potential *L. citreum* was isolated and analyzed for its biological potential, including survival ability in acidic and bile salt environments, antibacterial activity against infection-causing bacterial pathogens by different in vitro methods including time- and dose-based killing assay, minimum inhibitory concentration (MIC) and minimum bactericidal concentration (MBC), as well as competitive growth between LAB and bacterial pathogens.

## 2. Methods and Materials

### 2.1. Lactic Acid Bacteria Isolation (LAB) and Characterization

The whole rice plant was collected from an agriculture field in Division of Grassland and Forage, Cheonan, South Korea. The LAB were isolated using De Man Rogosa and Sharpe agar (MRS) and the identity was confirmed by bromocresol purple agar medium (BCP) [23,24]. APICH50 and AbiZym detection kits (MarcyI Etoile, France) were used to measure carbohydrate fermentation and extracellular enzyme production by selected strains, respectively. The 16S rRNA sequences of isolate were performed by Solgent Pvt Ltd. (Daejeon, Republic of Korea) and identified at the species by NCBI blast tool and the sequences were deposited to NCBI GeneBank (GeneBank: OL677067-OL677068).

### 2.2. Probiotic Potential of Selected LAB

Active fresh cultures (24 h) of isolated strains were tested in simulated gastrointestinal juices (gastric, duodenal and intestinal) [23,25]. A hydrophobicity assay was conducted [26] and the aggregation properties [27] of selected isolates were determined.

### 2.3. Antibacterial Activity Well Diffusion 

Isolates were grown in MRS broth for 48 h at 37 °C in a 1000 mL glass bottle with butyl stoppers. Cell free supernatant (CFS) was collected by centrifuge for 30 min at 4000× *g*, 4 °C and used to determine antibacterial activity. *S. aureus*, *E. faecalis*, *E. coli* and *P. aeruginosa* were obtained from KACC, Korea (Korean Agriculture Culture Collection) and cultured in nutrient broth (NB) for 24 h at 37 °C. Each pathogen at the density of 10^8^ CFU/mL was spread onto nutrient agar (NA) plates and made a well on it. The CFS (100 μL/well) was dispensed into the well which was made on NA and incubated at 37 °C for 48 h. The inhibitory activity was monitored.

### 2.4. Lyophilization of Cell-Free Supernatant (CFS) 

The isolates were grown in MRS broth at 37 °C for 48 h with mild shaking in an arbitrary shaker. The CFS was separated at 4000× *g*, 4 °C for 30 min and then the pH of the CFS was adjusted with 1N NaOH and filtered by different filter papers with different pour size. CFS was then lyophilized under pressure (less than 50 m Torr) using an Ilshin Biobase (Gyeonggi-do, Republic of Korea).

### 2.5. Antibacterial Activity by Dose- and Time-Based Killing Assay, Minimum Inhibitory Concentration (MIC) and Minimum Bactericidal Concentration (MBC)

The CFS was suspended in NB at the final concentration of 25 mg/mL. Then, the samples were serially diluted (twofold) with same broth from 25 mg/mL to 0.02 mg/mL. The diluted samples were then (100 μL/mL) transferred to microplate wells with triplicates and ten microliters of each pathogen was inoculated into each well. Blank wells with CFS from each concentration were also maintained. The experimental plates were incubated for 48 h at 37 °C. The samples from each well were taken at different incubatory intervals (12, 24, 36 and 48 h) and intensity of samples was measured at 600 nm. The absorbance values were then normalized with blanks [28]. The MIC and MBC were also identified [23,29].

### 2.6. Antibacterial Activity by Co-Culture Method

The competitive growth between pathogen and LAB was identified by co-culture method [30,31] with minor modifications. MRS and nutrient broth were used to cultivate LAB and pathogen, respectively. The bacterial cultures were grown at 37 °C for 24 h. The pellets were then collected by centrifugation at 4000× *g*, 4 °C for 30 min. Then, bacterial pellets were washed with PBS twice, and suspended in PBS. For co-culturing LAB with pathogen, MRS and nutrient broth was used at a 1:1 ratio. The control (monoculture) was grown at 37 °C with individual inoculations of LAB and pathogen. Sampling was performed at 12, 24 and 36 h. On MRS and NA agar plates, colonies were counted and compared with monocultures.

### 2.7. Statistical Analysis

Data obtained from the experiments were analyzed by SPSS16 version statistical software (SPSS Inc., Chicago, IL, USA). The experimental data were represented as the Mean ± STD. Statistics were considered significant when *p* values were less than 0.05.

## 3. Results

### 3.1. LAB Strain Isolation and Characterization

The selected isolates were characterized based on their primary antimicrobial activities and their ability to survive under gastrointestinal tract (GIT) conditions. These strains were Gram-positive, coccus shape, negative catalase and non-motile type (Table 1). These strains were identified as *Leuconostoc citreum* based on 16S rRNA sequence data. A number of carbohydrate substrates were fermented by both *L. citreum* KCC-57 and *L. citreum* KCC-58 including glucose, fructose, mannose, maltose, lactose, N-acetylglucosamine and gentiobiose, etc. (Appendix A). The strains also produce different types of extracellular enzymes such as alkaline phosphatases, acid phosphatases, Naphthol-AS-biphosphohydrolase, β-Glucosidase and α-Glucosidase (Appendix A).

### 3.2. Potential Probiotic Properties of Isolated Strains

#### 3.2.1. LAB in Simulated GIT Conditions

The strains should be able to survive in acidic and bile salt environments in order to be considered probiotics. Consequently, the survival capability of *L. citreum* KCC-57 and *L. citreum* KCC-58 in gastric juice with a pH of 2.5, duodenal juice with a pH of 5 and intestinal juice with a pH of 8 were evaluated. *L. citreum* KCC-57 and *L. citreum* KCC-58 survived in gastric juice at rates of 30.25% and 34.37%, respectively. Duodenal juice (*L. citreum* KCC-57: 41.98% vs. *L. citreum* KCC-58: 62.21%) and intestinal juice (*L. citreum* KCC-57: 87.55 vs. *L. citreum* KCC-58: 83.28%) survival rates were higher than gastric juice (Figure 1A).

#### 3.2.2. Hydrophobicity and Auto-Aggregation 

Selection of probiotic strains is dependent on their hydrophobicity and aggregation properties, which determine their adherence and colonization ability. The hydrophobicity properties of *L. citreum* KCC-57 and *L. citreum* KCC-58 varied with time in chloroform and xylene. Hydrophobicity properties of *L. citreum* KCC-57 were 21.97–63.37% in chloroform and 31.51–43.50% in xylene at various incubation times (30 min–180 min). *L. citreum* KCC-58 accounted for 29.36 to 71.60% in chloroform and 31.72–54.20% in xylene. All strains showed a much higher degree of hydrophobicity in chloroform than in xylene at 180 min (Figure 1B). A time-dependent auto-aggregation study showed that both strains had broad range of aggregation abilities. *L. citreum* KCC-57 showed gradual sedimentation increases in the bottom of tubes and claimed to reduce turbidity at the same time, whereas *L. citreum* KCC-58 showed lower sedimentation rates and higher turbidities. After 180 min of incubation, both strains displayed higher percentages of aggregation properties (Figure 1C).

### 3.3. The Antibacterial Activities of Selected Isolates L. citreum KCC-57 and L. citreum KCC-58 

#### 3.3.1. Agar Well Diffusion

*L. citreum* KCC-57 and *L. citreum* KCC-58 strains showed significant probiotic characteristics, and then we analyzed for their antibacterial activity against *S. aureus*, *P. aeruginosa*, *E. coli* and *E. faecalis. L. citreum* KCC-57 and *L. citreum* KCC-58 CFS showed a wide range of zones of inhibition against tested pathogens (Table 2). The CFS of *L. citreum* KCC-57 and KCC-58 showed strong inhibitory zones against *E. coli*, followed by *S. aureus*, *E. Faecalis* and *P. aeruginosa*.

#### 3.3.2. Time-Killing Assay

The effects of *L. citreum* KCC-57 (Figure 2) and *L. citreum* KCC-58 (Figure 3) at different concentrations were investigated after different incubation periods. Depending on the concentration of CFS used (0.02–25 mg/mL), different antibacterial activities were observed against pathogens between 12 and 48 h. Time-based treatment reduced pathogen growth compared with untreated pathogens. All pathogens were highly susceptible to CFS of both strains at 25 mg/mL. The pathogen’s growth was increased when the concentration of CFS was reduced with an extended period.

#### 3.3.3. MIC and MBC

There was a difference between *L. citreum* KCC-57 and *L. citreum* KCC-58 treatments in terms of MIC and MBC. KCC-57 had MIC values of 12.5 mg/mL for *E. coli*, *E. faecalis P. aeruginosa* and 25 mg/mL for *S. aureus.* A KCC-58 MIC value was between 6.25 mg/mL and 12.5 mg/mL for all tested pathogens, including *E. faecalis P. aeruginosa*, *S. aureus* and *E. coli.* Strong bactericidal activity (MBC) was noted against *E. coli* and *P. aeruginosa* at 25 mg/mL. No MBC values were observed for *E. faecalis* and *S. aureus*. KCC-58, at doses between 12.5 mg/mL and 25 mg/mL, showed strong bactericidal activity (MBC) against all tested pathogens (Table 3).

#### 3.3.4. Co-Culture of LAB with Pathogens

It was found that *L. citreum* KCC-57 and *L. citreum* KCC-58 have significant probiotic potential, as well as strong antibacterial activity. In modified broth (MRS-NB, 1:1), the LAB were grown competitively with pathogens. A co-culture between pathogens and LAB was shown to exert a strong inhibitory effect against pathogen growth via dominating LAB growth, as confirmed by growth of pathogens and LAB on nutrient agar and MRS agar, respectively. The ability of *L. citreum* KCC-57 and *L. citreum* KCC-58 to reduce pathogen growth could facilitate prevention of pathogen growth by increasing colonization and biofilm formation in the GIT (Table 4).

## 4. Discussion

Infectious bacteria are associated with a number of chronic and acute diseases, among them *E. faecalis S. aureus*, *P. aeruginosa* and *E. coli*, causing diarrhea [32], intestinal inflammation [33], colon cancer, oral cancer lesions [34] and urinary tract infections [35,36]. Antibiotics have made a significant contribution to medical science by being able to prevent pathogens from spreading, thereby decreasing death rates. Occasionally, antibiotics are not effective at controlling the spread of pathogens due to the emergence of antibiotic-resistant bacteria [3]. Therefore, effective strategies are needed to prevent and control such diseases. Researchers consider LAB to be the most potential probiotics because they possess a broad range of antimicrobial properties [37,38]. In the current study, two novel *L. citreum* KCC-57 and *L. citreum* KCC-58 were selected and characterized for their biological potential, including carbohydrate fermentation, enzyme production and survival in GIT juices, as well as antibacterial activities against *E. coli*, *E. faecalis*, *P. aeruginosa* and *S. aureus*. A variety of carbohydrate substrates could be fermented, and different enzymes were produced by both strains. The strains should have resistance to low pH and bile salt environments [39,40] in order to be considered probiotics. In this study, we found that both *L. citreum* KCC-57 and *L. citreum* KCC-58 survived significantly in simulated gastric juice (pH 2.5) without nutrients. Despite the fact that both strains showed a lower survival rate under this harsh condition, more than 30% of the strains were able to survive. Previous studies have been suggested that *Leuconostoc* spp. TBE-8 and P1 lose survival rates at lower pH (pH 3 and pH 4 in MRS broth), but a significant number of colonies can tolerate acidic pH in the presence of MRS containing nutrients. However, our strains *L. citreum* KCC-57 and *L. citreum* KCC-58 survived significantly in lower pH (pH 2.5) without nutrition (acidic pH in PBS) compared to previously reported *Leuconostoc* spp. [41,42]. Furthermore, the resistant capacity of *L. citreum* KCC-57 and *L. citreum* KCC-58 was found to be higher in duodenal and intestinal juices compared to gastric juice. It suggests that both *L. citreum* KCC-57 and *L. citreum* KCC-58 acquired significant stress tolerance and resistant capacities from gastric juice. Bile salt concentrations in the GIT range from 0.3 to 0.5% [43]. Probiotics with bile salt resistance are associated with the bile salt hydrolase enzyme, which hydrolyzes conjugated bile salts [44]. In the current study, it was confirmed that both strains were capable of surviving in a bile salt environment. In general, bile salts have detrimental effects that limit microbial growth. Overall, these strains gained tolerance in their stomach juice and improved their ability in subsequent unfavorable conditions, referred to as the cross-protective stress response [45], before reaching the intestinal wall and re-establishing the gut microbiome and inhibiting undesirable microbial growth [23]. *Leuconostoc sp.* showed better tolerance when bile salt concentrations were less than 0.4% [41]. An adhesion assay is used to assess bacteria’s ability to adhere to epithelia in the gastrointestinal tract [46,47]. Furthermore, auto-aggregation contributes to biofilm development that increases intestinal colonization and prevents pathogen adhesion. In the present study, *L. citreum* KCC-57 and *L. citreum* KCC-58 showed significant hydrophobicity characteristics in chloroform and xylene. Both strains expressed higher hydrophobicity properties in chloroform than in xylene. Additionally, these strains exhibited considerable levels of auto-aggregation in a time-dependent manner, confirming that selected isolates could be able to interact with mucous or epithelial cells which help to inhibit adhesion of pathogens [48]. LAB have antimicrobial activity directly associated with production of organic acids such as lactic acid, acetic acid, propionic acid, hydrogen peroxide, bacteriocins and peptides [49,50]. *L. citreum* KCC-57 and *L. citreum* KCC-58 have been shown to have strong antibacterial activity against *E. coli*, *P. aeruginosa*, *S. aeruginosa* and *E. faecalis* as determined by agar well diffusion, microdilution, MIC/MBC, time- and dose-dependent killing assay and dual culture methods. Several studies indicated that the *Lactobacillus* sp. and *Leuconostoc* sp. possessed strong antimicrobial activity via the production of secondary metabolites, nutrient utilization competition, inhibition of adhesion of bacterial to the intestinal mucosa and an improved immunological response [37,38,51]. CFS of *L. citreum* KCC-57 and *L. citreum* KCC-58 demonstrated broad antibacterial activity against all tested pathogens. Inhibitory activities have varied between strains *L. citreum* KCC-57 and *L. citreum* KCC-58. An MIC/MBC and time-/dose-dependent study also supported the antibacterial activity of *L. citreum* KCC-57 and *L. citreum* KCC-58. The maximum concentration of CFS used in this study (25 mg/mL) completely eradicated the growth of pathogens even after 48 h incubation. Finally, the competitive growth study between LAB and pathogens demonstrated that the LAB predominantly dominate the pathogen growth in the customized media. All the methods of antibacterial study suggested that *L. citreum* KCC-57 and *L. citreum* KCC-58 possessed strong antagonistic activity against infection-causing pathogens.

## 5. Conclusions

*L. citreum* KCC-57 and *L. citreum* KCC-58 were isolated from rice plants and investigated for their probiotic potential and antibacterial activity. These strains displayed significant survival rates under harsh simulated gastrointestinal conditions and showed good hydrophobicity as well as auto-aggregation properties, which makes them potential probiotic candidates. Additionally, both *L. citreum* KCC-57 and *L. citreum* KCC-58 exhibited strong antagonistic activity at different ranges. Co-culture analysis revealed that KCC-57 and KCC-58 inhibit pathogen growth through competitive inhibition, suggesting that they can be used as bio-therapeutic antibacterial agents instead of antibiotics. One of the advantages of the reported strains is its strong antagonistic activity against *E. coli*, *P. aeruginosa*, *S. aeruginosa* and *E. faecalis*. The disadvantage of the strains is reduced survival rates at pH 2.5. Therefore, these strains must be subjected to gastrointestinal tract conditions in order to determine the true survival rates and other biological potential in in vivo animal models.

## Figures and Tables

**Figure 1 microorganisms-11-00469-f001:**
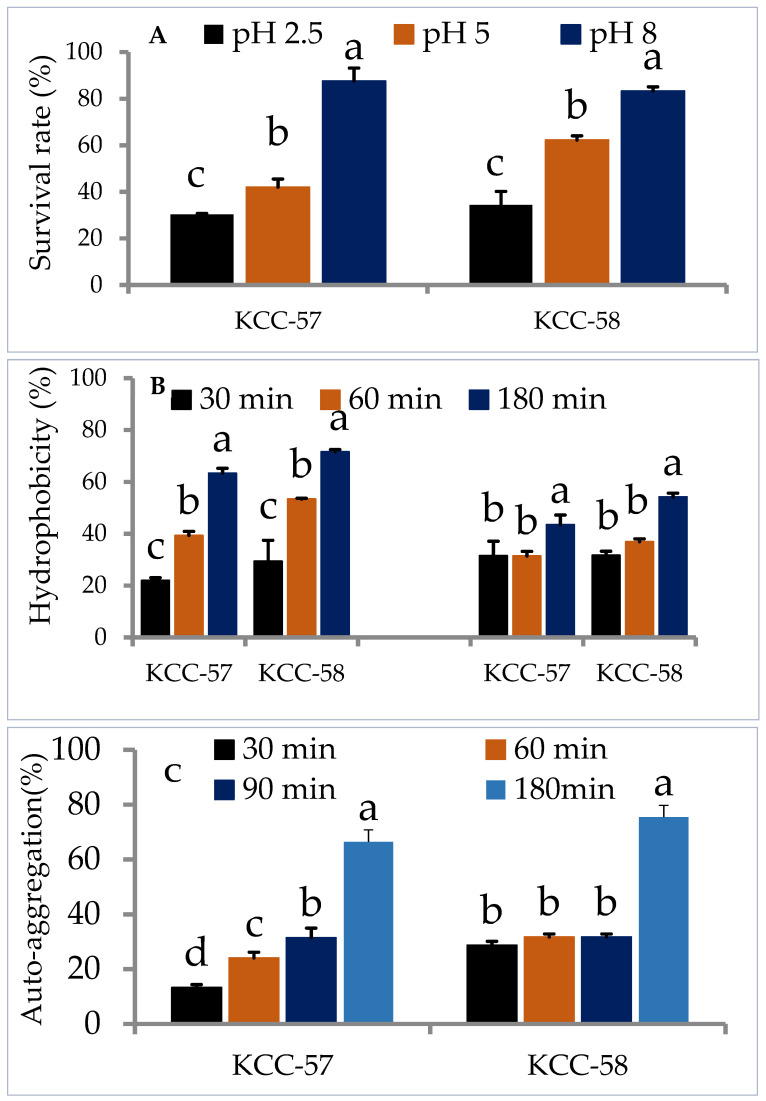
Probiotic characteristics of *L. citreum* KCC-57 and *L. citreum* KCC-58. (**A**) Survival ability of KCC-57 and KCC-58 in simulated gastrointestinal juices, (**B**) hydrophobicity properties of KCC-57 and KCC-58 in different hydrocarbons, (**C**) auto-aggregation property of KCC-57 and KCC-58 in a time-based incubation. The data are represented as the mean ± STD of six replicates. Different alphabets within the figure indicate significant differences between experimental data at the 0.05 level.

**Figure 2 microorganisms-11-00469-f002:**
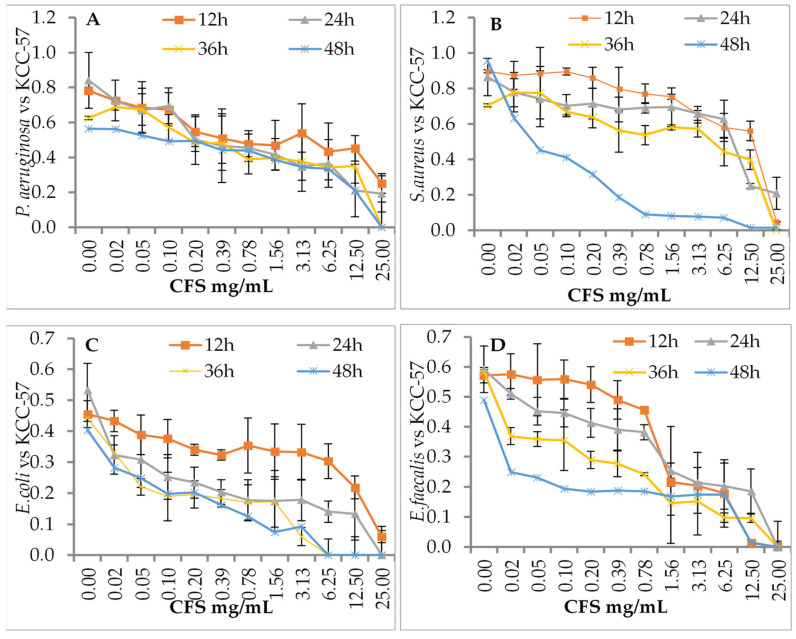
Antibacterial activity of CFS of KCC-57 by dose- and time-dependent killing assay method. (**A**) CFS vs. *P. aeruginosa*, (**B**) CFS vs. *S. aureus*, (***C**)* CFS vs. *E. coli*, (**D**) CFS vs. *E. faecalis.* The data are expressed as the mean ± STD (n = 3).

**Figure 3 microorganisms-11-00469-f003:**
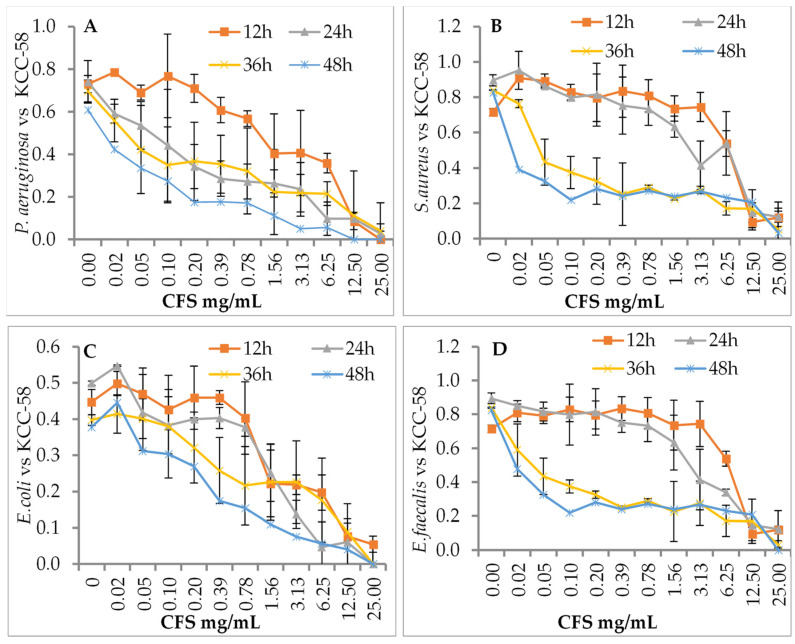
Antibacterial activity of CFS of KCC-58 by dose- and time-dependent killing assays. (**A**) CFS vs. *P. aeruginosa*, (**B**) CFS vs. *S. aureus*, (***C**)* CFS vs. *E. coli*, (**D**) CFS vs. *E. faecalis.* The data are expressed as the mean ± STD (n = 3).

**Table 1 microorganisms-11-00469-t001:** Morphophysiological features of identified isolates of *L. citreum* KCC-57 and *L. citreum* KCC-58.

Code	Stain	Motile	Shape	Catalase	Growth Condition	Strain	Species
**KCC-57**	Positive	Non	Cocci	Negative	Facultative anaerobic	*Leuconostoc*	*citreum*
**KCC-58**	Positive	Non	Cocci	Negative	Facultative anaerobic	*Leuconostoc*	*citreum*

**Table 2 microorganisms-11-00469-t002:** Antibacterial activity of CFS of KCC-57 and KCC-58 by well diffusion method.

Code	Species Name	Zone of Inhibition (mm)			
		*E. coli*	*S. aureus*	*P. aeruginosa*	*E. faecalis*
**KCC-57**	*L. citreum*	34.3 ± 1.2	22.0 ± 2.9	17.5 ± 0.35	18.5 ± 1.06
**KCC-58**	*L. citreum*	32.6 ± 2.4	27.3 ± 3.2	18.5 ± 0.35	22.5 ± 1.77

CFS: Cell free supernatant.

**Table 3 microorganisms-11-00469-t003:** Minimum inhibitory concentration (MIC) and minimum bactericidal concentration (MBC) of CFS of KCC-57 and KCC-58.

Pathogens	KCC-57		KCC-58	
	MIC (mg/mL)	MBC (mg/mL)	MIC (mg/mL)	MBC (mg/mL)
*E. coli*	12.5	25	12.5	25
*E. faecalis*	12.5	-	6.25	12.5
*P. aeruginosa*	12.5	25	12.5	25
*S. aureus*	25	-	12.5	25

**Table 4 microorganisms-11-00469-t004:** Antibacterial activity of live *L. citreum* against various pathogens using co-culture method.

Groups	Growth on MRA Agar	Growth on NA	Groups	Growth on MRA Agar	Growth on NA	Groups	Growth on NA
**Bacterial growth (10^7^ CFU/mL) after 12 h**							
**KCC-57 alone**	5.92 ± 0.68		KCC-58 alone	5.72 ± 0.42			
**KCC-57 ± SA**	2.52 ± 0.69	0.38 ± 0.12	KCC-58 + SA	4.32 ± 1.30	0.776 ± 0.07	SA alone	28.0 ± 0.56
**KCC-57 ± PA**	5.10 ± 0.66	0.78 ± 0.02	KCC-58 + PA	4.56 ± 0.05	0.736 ± 0.03	PA alone	28.4 ± 1.98
**KCC-57 ± EC**	5.80 ± 0.14	0.02 ± 0.02	KCC-58EC	2.60 ± 1.13	0.028 ± 0.02	EC alone	29.5 ± 2.80
**KCC-57 ± EF**	3.40 ± 0.56	0.18 ± 0.02	KCC-58 + EF	3.60 ± 0.28	0.144 ± 0.01	EF alone	23.4 ± 0.07
**Bacterial growth (10^7^ CFU/mL) after 24 h**							
**KCC-57 alone**	6.92 ± 0.19		KCC-58 alone	7.72 ± 0.65			
**KCC-57 + SA**	1.16 ± 0.03	0.564 ± 0.10	KCC-58 + SA	2.46 ± 0.05	0.10 ± 0.01	SA alone	54.0 ± 6.29
**KCC-57 + PA**	1.02 ± 0.01	0.288 ± 0.01	KCC-58 + PA	3.32 ± 0.07	0.23 ± 0.04	PA alone	47.3 ± 1.81
**KCC-57 + EC**	3.80 ± 0.14	0.104 ± 0.01	KCC-58 + EC	2.70 ± 0.01	1.36 ± 0.07	EC alone	35.6 ± 2.26
**KCC-57 + EF**	3.50 ± 0.35	0.288 ± 0.09	KCC-58 + EF	3.20 ± 0.42	0.14 ± 0.01	EF alone	38.7 ± 3.40
**Bacterial growth (10^7^ CFU/mL) after 36 h**							
**KCC-57 alone**	6.66 ± 0.24		KCC-58 alone	6.49 ± 0.46			
**KCC-57 + SA**	2.21 ± 0.50	0.384 ± 0.12	KCC-58 + SA	1.59 ±0.52	0.676 ± 0.07	SA alone	48.8 ± 2.95
**KCC-57 + PA**	3.34 ± 0.08	0.528 ± 0.34	KCC-58 + PA	1.64 ± 0.15	1.080 ± 0.09	PA alone	33.8 ± 2.07
**KCC-57 + EC**	3.00 ± 0.42	0.116 ± 0.03	KCC-58 + EC	2.50 ± 0.21	0.064 ± 0.01	EC alone	37.7 ± 2.76
**KCC-57 + EF**	2.30± 0.21	0.120 ± 0.00	KCC-58 + EF	2.10 ± 0.49	0.120 ± 0.01	EF alone	38.5 ± 5.03

SA-*Staphylococcus aureus*; PA-*Pseudomonas aeruginosa*; EC-*Escherichia coli*, and EF-*Enterococcus faecalis*. Co-culture: both LAB and pathogen were grown in modified media (MRS: NA 1:1); monoculture: both LAB and pathogen were grown in modified media; both cultures were incubated at 37 °C for 48 h. LAB and pathogen colonies were monitored using MRS and NA plates, respectively. The data were then compared with monoculture. Data are presented as mean ± standard deviation (n = 3).

## Data Availability

Please contact the corresponding author if you would like to access the experimental data.

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
