# Peer review of "A Novel Strain of Probiotic Leuconostoc citreum Inhibits Infection-Causing Bacterial Pathogens"

_microorganisms, 2023, doi:10.3390/microorganisms11020469_

Round 1
Reviewer 1 Report
The aims of the study must be clearly stated both in the abstract and at the end of the introduction.
"These strains were identified as Leuconostoc citreum based on 16srRNA sequence data." Which strains? Please give details on the methodology. Please deposit the DNA sequences and give their accession numbers.
In the Conclusions section, please highlight whether the aims of the study were achieved.
And why are these two Leuconostoc species different from other previously published probiotic strains? What is essentially the novelty of this study?
Minor comments:
Please correct some typos along the manuscript, for example:
-in the abstract: "16srRNA"; "Pseudimonas", etc.
-lines 126-127: "minimum inhibitory concentration (MBC) and minimum bactericidal concentration (MIC)"
-Table 2: "E. Faecalis"
Author Response
Reviewer 1
We greatly appreciate your valuable comments and suggestions regarding our submitted article, and these suggestions will help to improve both the quality and presentation of the manuscript. We have revised the entire manuscript based on the reviewer's comments. All changes made to the manuscript have been highlighted in red. We hope that the revised manuscript could be suitable for acceptance in your journal.
- These strains were identified as Leuconostoc citreum based on 16srRNA sequence data." Which strains? Please give details on the methodology. Please deposit the DNA sequences and give their accession numbers.
Thank you for your kind suggestions and recommendation. We have provided a GeneBank number for reported Leuconostoc citreum. The isolated bacterial 16S rRNA sequence was analyzed by Solgent Pvt Ltd (Daejeon, Korea) and identified at the species by NCBI blast tool and sequences were submitted to the NCBI GeneBank (GeneBank: OL677067- OL677068)
- In the Conclusions section, please highlight whether the aims of the study were achieved.
The conclusion has been revised as L. citreum KCC-57 and L. citreum KCC-58 were isolated from rice plants and investigated for their probiotic potential and antibacterial activity. These strains displayed significant survival rates under harsh simulated gastrointestinal conditions and showed good hydrophobicity as well as auto-aggregation properties, which is making them potential probiotic candidates. Additionally, both L. citreum KCC-57 and L. citreum KCC-58 exhibited strong antagonistic activity at different ranges. Co-culture analysis revealed that KCC-57 and KCC-58 inhibit pathogen growth through competitive inhibition, suggesting that they can be used as bio-therapeutic antibacterial agents instead of antibiotics. One of the advantages of the reported strains is its strong antagonistic activity against E.coli, P. aeruginosa, S. aeruginosa, and E. faecalis. The disadvantage of the strains is reduced survival rates at pH 2.5. Therefore, these strains must be subjected to gastrointestinal tract conditions for determining the true survival rates and other biological potential in in vivo animal models.
- And why are these two Leuconostoc species different from other previously published probiotic strains? What is essentially the novelty of this study?
There are a limited number of research articles available about the biological potential of Leuconostoc citrum. We have therefore experimented with potential Leuconostoc citrum strains. Currently, the reported strains were less likely to survive in the simulated gastric juice environment. However, compared to other Leuconostoc species, these strains exhibit significant survival rates at lower pH levels, in particular pH 2.5 without nutrients. Previously reported Leuconostoc species also had different survival rates at pH 4 (lower pH in MRS broth) when nutrients were present. Now we have interpreted and compared our reported strains results to other published data.
- Please correct some typos along the manuscript, for example:
Thanks for sharing your valuable information. The whole manuscript has been reviewed and all errors and mistakes have been corrected.
- in the abstract: "16srRNA"; "Pseudimonas", etc.
Thank you very much. It has been revised as 16S rRNA and Pseudomonas aeruginosa throughout the manuscript.
- lines 126-127: "minimum inhibitory concentration (MBC) and minimum bactericidal concentration (MIC)"
Thank you a lot. The error has been revised as minimum bactericidal concentration (MIC)"
- Table 2: "E. Faecalis"
Thank you very much. It has been revised as E.faecalis.

Reviewer 2 Report
Dear Authors!
My comments are in attached file.

Author Response
Reviewer 2
Manuscript by Muthusamy et al. “A Novel Strain of Probiotic Leuconostoc Citrum Inhibits Infection-Causing Bacterial Pathogens” is dedicated to the characterization of new probiotic strains of Leuconostoc Citrum. The work contains new data and attempts to analyze the possible use of these strains in practice. However, I have a number of significant comments that need to be clarified.
We greatly appreciate your valuable comments and suggestions regarding our submitted article, and these suggestions will help to improve both the quality and presentation of the manuscript. We have revised the entire manuscript based on the reviewer's comments. All changes made to the manuscript have been highlighted in red. We hope that the revised manuscript could be suitable for acceptance in your journal.
- Major criticism General note - the text should be carefully formatted and edited for errors, typos, lack of spaces between words, enter all the necessary abbreviations.
Thanks for sharing your valuable suggestions. The article has been updated, and abbreviations have been explained.
- INTRODUCTION The first paragraph of the Introduction is too verbose and should be shortened by focusing on the problem of finding new probiotics. Perhaps this text should be moved to the discussion.
We agree with the reviewer's comments. The introduction section has been shortened in accordance with your suggestion and contains clear statements about infection causing death worldwide and the importance of prevention methods and the development of new strategies.
The global population rate and microbial pathogen associated infections have been increasing continuously for decades. Millions of lives are lost every year due to microbial infections. Antimicrobial resistance (AMR) is the primary cause of microbial infections. It is one of the biggest threats to human health in 21st century because pathogens are becoming less susceptible to drug treatments. AMR is predicted to kill ten million people by 2050 (O'Neill, 2016; O’Neill, 2014). According to the world health organization (WHO) and other researchers, AMR is an emergency issue that requires key strategies to control it globally (Prestinaci et al., 2015; Prevention, Antibiotic resistance threats in the United States; WHO, 2020). There were 4.9 million deaths in 2019 from bacterial infections, of which 1.27 million were caused by antimicrobial resistance. At all ages, Western sub-Saharan Africa had the highest death rates due to AMR. Among the pathogens E.coli, A. baumannii K. penumoniae P. aeruginosa and S. aureus are associated with antimicrobial resistance. 929000 deaths were caused by these pathogens in 2019 and 3.57 million in 2020. Over 100000 deaths have been associated with methicillin-resistant S.aureus in 2019(Murray et al., 2022).
In order to control AMR pathogens, effective antibiotics must be invented. Since, antibiotics have become less effective due to microbial resistance. Therefore, treatment has become more challenging. There is a need for emerging and alternative strategies to protect valuable life from AMR pathogens. According to recent research, probiotics are the most effective tool for controlling AMR pathogens. Probiotic bacteria such as Lactobacillus and Bifidobacterium have a variety of biological applications. (Janiszewska-Turak et al., 2021). Lactic acid bacteria (LAB) produce a variety of secondary metabolites that are potential antimicrobials, antioxidants, and other biological agents (Aponte et al., 2020; Muthusamy et al., 2020; Soundharrajan et al., 2020). The secondary metabolites of LAB inhibit pathogen growth via multiple molecular mechanisms (Byakika et al., 2019; Kang et al., 2018; Kim et al., 2021; Vieco-Saiz et al., 2019; Zhang et al., 2020). LAB can also prevent the attachment of pathogenic microbes to the epithelial cells by competing with pathogens and reducing colonization of pathogens, thereby reducing infection. (Garriga et al., 2015; Guan et al., 2020; Zawistowska-Rojek et al., 2022). Therefore, we intended to use Leuconostoc citrum from rice plants to control various pathogens in this regard. L. citrum a dextran-producing species and is believed to have significant probiotic potential due to its putative plasmid-endcoded cell wall anchored protein, which contains five mucus binding domains essential for colonizing the digestive system. As well as producing various antimicrobial peptides (Iosca et al., 2022; Pujato et al., 2014; Woo et al., 2021), it is also capable of killing cancer cells (Holland, 2011). Hence, in the present study, potential L. Citrums was isolated and analyzed for their biological potential, including survival ability in acidic and bile salt environments, antibacterial activity against infection-causing bacterial pathogens by different in vitro methods including time and dose based killing assay, minimum inhibitory concentration (MIC) and minimum bactericidal concentration (MBC), as well as competitive growth between LAB and bacterial pathogens
- METHODS : The work contains many methodological inaccuracies that must be eliminated: The authors write that the antibacterial activity was analyzed at 32 °C: “2.3. Antibacterial activity well diffusion Isolates were grown in MRS broth for 48h at 32°C in a 1000mL glass bottle with butyl stoppers”. Why studies were not conducted at 37 °C - a temperature close to the internal environment of the human body? Can these species potentially grow and reproduce at 37°C? 2.
We understand the reviewer's concern about the temperature of the LAB for growth. The selected strains KCC-57 and KCC-58 can grow in MRS medium at 32°C to 37°C in orbital shakers at 125rpm. We have revised the growth temperature for KCC-57 and KCC-58 as Isolates were grown in MRS broth for 48h at 37°C in a 1000mL glass bottle with butyl stoppers. Cell free supernatant (CFS) was collected by centrifuge for 30minutes at 4000g, 4°C and used to determine antibacterial activity. S. aureus, E. faecalis, E. coli and P. aeruginosa were obtained from KACC, Korea (Korean Agriculture Culture Collection) and cultured in Nutrient broth (NB) for 24h at 37°C. Each pathogen at the density of 108CFU/mL was spread onto nutrient agar (NA) plats and made the well on it. The CFS (100μL/well) was dispensed into well, which made on NA and incubated at 37°C for 48h. The inhibitory activity was monitored.
- Why was alkalization of the medium performed during lyophilization of biomass? (2.4. Lyophilization of cell-free supernatant (CFS).
The natural cell-free supernatant has an acidic pH (pH<4). The growth of pathogens would be prevented in an acidic environment. If we use natural CFS to determine antibacterial activity, we cannot predict if the inhibition is caused by its acidic nature or secondary metabolites. Therefore, we used neutralized CFS for an antibacterial study.
- 144 the bacterial cultures were grown at appropriate temperatures for 24 hours. – at what temperature were the strains cultivated?
This has now been changed to 37°C for 24 hours in order to grow the bacterial cultures.
- RESULTS Table 1. Physiochemical features – it would be more correct to call the properties morphophysiological. But in my opinion, Table 1 does not contain significant information; its content can be briefly described in the text.
We have agreed with the reviewer's comment and modified the same as Morphophysiolgical features of identified isolates of L. citreum- KCC-57 and L. citreum KCC-58
- Figure 2- 3 – low resolution
The quality of figures 2 and 3 has been improved based on reviewer recommendations.
- DISCUSSION If the survival of the studied strains in the stomach is about 30% of the initial one, what will it be like in the lower parts of the GIT, considering that at intestinal pH (pH 5) it decreases by more than 2 times? Is it possible to extrapolate what would be the effectiveness of probiotics in a model organism? In addition, those parts that duplicate the presentation of the results should be rewritten, with an emphasis on comparing the results obtained with similar ones, known in the literature and comments on the advantages of the studied strains compared to other probiotic species.
Yes, we agree with the reviewer's comment. Our strains' survival rates were lower in the simulated gastric juice environment. As compared to other Leuconostoc species, our reported strains are significantly more resistant to lower pH, in particular pH 2.5 without nutrients. As previously reported, Leuconostoc species have different survival rates when nutrients are present (lower pH in MRS broth). Our strain results have now been interpreted and compared to other published data.
In this study, we found that both L. citreum KCC-57 and L. citreum KCC-58 survived significantly in simulated gastric juice (pH 2.5) without nutrients. Despite the fact that both strains showed a lower survival rate under this harsh condition, more than 30% of the strains were able to survive. Previous studies have been suggested that Leuconostoc spp TBE-8 and P1 lose survival rates at lower pH (pH 3 and pH 4 in MRS), but a significant number of colonies can tolerate acidic pH in the presence of MRS containing nutrients. But, our strains L. citreum KCC-57 and L. citreum KCC-58 survived significantly in lower pH (pH 2.5) without nutrition (Acidic pH in PBS) compared to previously rreported Leuconostocs pp [40,41]. Furthermore, the resistant capacity of L. citreum KCC-57 and L. citreum KCC-58 was found to be higher in duodenal and intestinal juices compared to gastric juice. It suggests that both L. citreum KCC-57 and L. citreum KCC-58 acquired significant stress tolerance and resistant capacities from gastric juice. Bile salt concentrations in the GIT range from 0.3 to 0.5% [42]. Probiotics with bile salt resistance are associated with the bile salt hydrolase enzyme, which hydrolyzes conjugated bile salts [43]. In the current study, it was confirmed that both strains were capable of surviving in a bile salt environment. In genaral, bile salts have detrimental effects that limit microbial growth. Overall, these strains gained tolerance in their stomach juice and improved their ability in subsequent unfavorable conditions, referred to as the cross-protective stress response (Buriti et al., 2010), before reaching the intestinal wall and re-establishing the gut microbiome and inhibiting undesirable microbial growth[23]. Leuconostoc sp showed better tolerance when bile salt concentrations were less than 0.4% [44].
- CONCLUSION it’s a summary of the results. It should be supplemented with proposals on the potential use of LAB strains in practice, taking into account all their advantages and disadvantages.
Yes we have revised as L. citreum KCC-57 and L. citreum KCC-58 were isolated from rice plants and investigated for their probiotic potential and antibacterial activity. These strains displayed significant survival rates under harsh simulated gastrointestinal conditions and showed good hydrophobicity as well as auto-aggregation properties, which is making them potential probiotic candidates. Additionally, both L. citreum KCC-57 and L. citreum KCC-58 exhibited strong antagonistic activity at different ranges. Co-culture analysis revealed that KCC-57 and KCC-58 inhibit pathogen growth through competitive inhibition, suggesting that they can be used as bio-therapeutic antibacterial agents instead of antibiotics. One of the advantages of the reported strains is its strong antagonistic activity against E.coli, P. aeruginosa, S. aeruginosa, and E. faecalis. The disadvantage of the strains is reduced survival rates at pH 2.5. Therefore, these strains must be subjected to gastrointestinal tract conditions for determining the true survival rates and other biological potential in in vivo animal models.
- Minor remarks Lactic acid bacteria (LAB) should be kept as an abbreviation at the beginning of the article in vitro – should be by italic MRS agar, BCP agar – to decipher
Yes, we have incorporated the reviewer's comment and provided the abbreviations at the beginning.
These lines have been revised as The LABs were isolated using De Man Rogosa and Sharpe agar (MRS) and the identity was confirmed by bromocresol purple agar medium (BCP)
- l118 2.4. Lyophilization of cell-free supernatant (CFS) – reduction has already been introduced
Yes, it has been removed as suggested by the reviewer.
- 110 - bottle with butyl stoppers. Cell free supernatant (CFS) L 127 minimum bactericidal concentration (MIC) may be minimum inhibitory concentration?
Thanks a lot. Yes, it is the minimum inhibitory concentration. Now it has been revised throughout the manuscript.
- References in the text and a list of references should be arranged according to the rules of the journal.
Thanks for your suggestion. Text and list references have been revised according to journal format
- In general, I believe that the article can be considered after significant revision.
It is a pleasure to hear such positive comments about our research article.

Round 2
Reviewer 1 Report
The authors have satisfactorily revised their manuscript following the suggestions of both reviewers.
Reviewer 2 Report
Dear Authors!
MS looks much more better. The authors certainly did a great job.
I am satisfied with the responses to my comments. The only comment is that in Figure 2 B and C, the numbers along the y-axis and the caption are superimposed on each other. Taking into account this remark, the article can be accepted for publication.